# Characteristics of Vegetation Resistance Variation in Muddy Water Flows

**Xiaolei Zhang \*, Yu Zhu, Haoran Wu, Zhengzheng Bi and Zhiheng Xu**

Institute of Water Conservancy, North China University of Water Resources and Electric Power, Zhengzhou 450046, China; zy0900501@163.com (Y.Z.); wuhaorande@outlook.com (H.W.); bzz97@foxmail.com (Z.B.); 18237801369@163.com (Z.X.)
\* Correspondence: zxl1334@163.com

**Abstract:** The shoal area of the lower Yellow River in China is not flooded with water during the dry season, so various plants can grow. When floods overflow the plains in the flood season, the complexity of water resistance is increased due to the resistance to water flow by vegetation, which directly affects flood discharge in the beach area. The drag force coefficient ($C_D$), Manning's roughness coefficient ($n$), and Darcy-Weisbach resistance coefficient ($f$) are commonly used to characterize vegetation drag force. Such studies are commonly conducted in clear water, but flood water in the lower Yellow River is generally muddy. In order to study the effect of the same sediment content and different sedimentation thicknesses on the resistance of muddy waters containing vegetation, this study conducted experiments in a flume (length = 28 m, width = 0.5 m, and height = 0.5 m) under different deposition thicknesses. The results showed that the vegetation drag force coefficient ($C_D$), vegetation roughness ($n_b$), and Darcy-Weisbach drag coefficient ($f$) all decreased logarithmically with increasing Reynolds number ($Re$) and Froude number ($Fr$). When $Re > 30,000$, under the conditions of different siltation thicknesses of vegetation, the vegetation roughness tended to stabilize near its minimum value. When the Reynolds number of the water flow is large ($Re > 20,000$), the variation of the Darcy-Weisbach drag coefficient $f$ slows down with the Reynolds number $Re$. Logarithmic functions were established for the above resistance coefficients and flow coefficients, and the corresponding correlation coefficients were high, indicating that the conclusions were reliable.

**Keywords:** logarithmic fitting; muddy water containing vegetation; water resistance; water flume test

## 1. Introduction

Due to human activities, beaches are often interrupted by various kinds of vegetation and water-blocking structures, which increase surface roughness along beaches relative to the main trough [1–5], which directly affects the evolution of floodplains in beach areas. Floodplains are generally flooded only during the flood season, so when dry, various types of vegetation can grow in tidal areas. Beach flow resistance is closely related to the distribution and growth of beach vegetation and the distribution of various water-blocking structures [6–10]. The presence of vegetation has a particularly important effect on altering the movement characteristics of sand-carrying flow in beach areas as opposed to open channels, and the resistance of beach areas will be significantly increased by vegetation [11–15]. Similar processes occur in muddy areas, such as the Yellow River basin, so it is necessary to study the resistance characteristics of muddy substrates containing vegetation, which will provide a basis for understanding flood discharge and managing the ecological protection of muddy areas during flood seasons.

Vegetation can be divided into rigid vegetation and flexible vegetation. Vegetation with sufficient rigidity will not be deformed by water motion. Generally speaking, the structure of water flowing through a single plant can be regarded as the flow around a cylinder [16]. Huang et al. [17] used metal rods with different diameters to simulate

the water flow resistance caused by planting trees in beach areas, and on this basis, they demonstrated that the resistance of the model and prototype were similar. Their work provided a guide for the design of water flow resistance experiments investigating the effects of vegetation. Stone and Shen [18] used rigid cylindrical rods to study the effect of vegetation diameter and density on resistance at different water depths and found that the combined action of the above three factors determined resistance. James et al. [19] showed that vegetation roughness changed linearly with water depth when vegetation was in a non-submerged state. To explore the change in vegetation roughness with water depth, Wu et al. [20] used a variable slope flume river model with horsehair instead of vegetation. They found that, in the case of submerged vegetation, the vegetative roughness coefficient is negatively correlated with the depth of the water. However, the roughness increased gradually with the depth of the water until the vegetation was completely submerged, at which point it began to decrease before eventually stabilizing and becoming constant. Järvelä [12] carried out flume experiments with willow branches, seagrass, and sedge. The resistance coefficients were calculated as the change in the water head measured by a pressure difference sensor, from which the influence rules of different plant types, densities, arrangements, and water depths on the resistance were obtained. Their results showed that the resistance coefficient of non-submerged willow branches increased linearly with water depth, and under the same arrangement, the resistance coefficients of branches with subbranches were 2~3 times those without. Lu et al. [21] studied the change rules of vegetation resistance coefficients under different vegetation densities, flow conditions, and water depths. They adopted the vegetation hydraulic radius as the characteristic length to represent vegetation and the Reynolds number to represent flow conditions when examining the relationship between the vegetation resistance coefficient and flow. Li and Shen [22] conducted an experiment on the influence of plants on water flow resistance and showed that the arrangement of vegetation had a significant effect. With the same vegetation density, a plum blossom crisscrossing vegetation arrangement had greater water flow resistance than the front and back parallel vegetation arrangements. Kouwen and Li [23] proposed that the factor affecting the boundary roughness was the bending degree of plants, and the greater the resistance coefficient, the greater the bending moment force exerted on plants. Therefore, both plant stiffness and density affect boundary roughness in water flow.

In summary, the research to date on resistance characteristics of water flow with vegetation has been relatively comprehensive, and the interaction between water flow and vegetation has been clearly established in terms of stiffness, flexibility, and vegetation. Several characteristic parameters, such as the drag coefficient, roughness coefficient, and Darcy-Weisbach resistance coefficient, have been utilized. However, although the above studies have been relatively comprehensive, they have all been based on clear water conditions. The flood of the lower Yellow River is generally muddy water with a relatively high sediment content, which further complicates the flow movement problem. On the one hand, the flow of sand can shape the boundary, which in turn affects the flow of water. Furthermore, the water flow carries sediment, and the presence of sediment in turn changes the physical properties and turbulent structure of the water flow, which in turn affects its energy loss, flow velocity distribution, and sand content distribution, and then affects the drag coefficient, roughness coefficient, and Darcy-Weisbach resistance coefficient. Moreover, the velocity of floodplain flood waters slows when flowing through intertidal areas, facilitating siltation and changing the topography of the intertidal area, which directly affects the evolution of floodplain floods in the intertidal area. The existing semi-empirical and semi-theoretical resistance formulas do not accurately describe the change laws of vegetation resistance in the beach areas of the lower Yellow River because models generally regard beach resistance as a fixed value in two-dimensional water and sediment that does not change with the water and sediment conditions. Furthermore, in the previous physical models, the beach morphology remained unchanged because they did not consider the deposition of sediment in the vegetation area. Therefore, in order to

obtain accurate resistance characteristics for floodplain floods in the lower Yellow River, this investigation carried out a flume experiment with muddy water containing vegetation.

## 2. Materials and Methods

### 2.1. Design of the Experimental Installation

The basic theory of open channel flow was used in this study. Many tests on the resistance characteristics of water flow with vegetation have used rectangular flumes; therefore, to ensure test comparability and the reliability of the results, this test used a rectangular flume with a variable slope. The bottom slope could be adjusted between 0 and 1%. The structure of the test tank is shown in Figure 1. According to the purpose of the test, the following design points were determined:

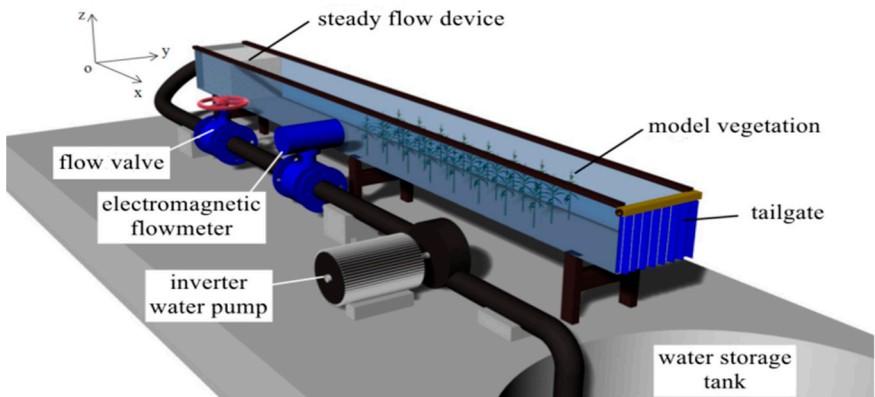

**Figure 1.** Experimental flume setup.

(1) The tank was 28 m in length, 0.5 m in width, and 0.5 m in height. Its side walls and bottom surfaces were made of rectangular sections of smooth tempered glass. In order to eliminate waves and stabilize water flow, the inlet of the water tank was the same size as the cross section of the tank. A 30-cm-long flow stabilizing device was used, composed of a number of PVC pipes placed inside the water tank. A 2.4-m-long, 0.5-m-wide, and 5-mm-thick PVC board was laid in the middle of the tank. Round holes are drilled into the board to fix the model plants.

(2) The model plants were typical maize vegetation found in the floodplain area, with a total vegetation height ($H_v$) of 21.5 cm, a lower main stem height of 5.5 cm, and a diameter of 0.15 cm. The upper branches and leaves had an average height of 12 cm. The length of the test control section was 1.4 m, and the width was the distance between the two inner walls of the water tank. The model plants were set in a rectangular arrangement (Figure 2), with a spacing of 10 cm between front and rear and left and right. The planting area per unit area was $9.091 \times 10^{-3}$ m², and the plant density was 110 plants/m².

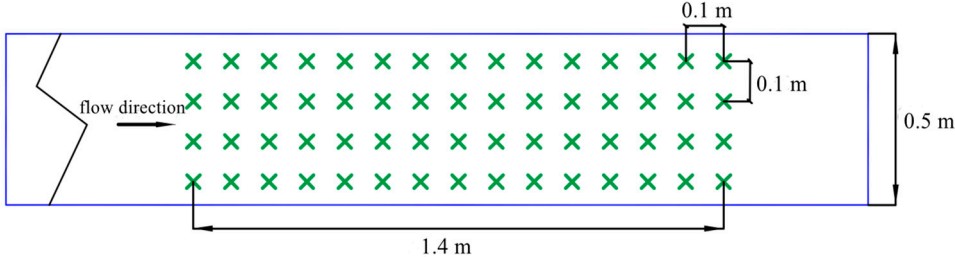

**Figure 2.** Schematic diagram of vegetation arrangement.

(3) The upper part of the tank had a movable bracket to facilitate the placement of measuring instruments. A tail door was set at the tail end of the tank to ensure that the water depth and flow were stable at a constant value. In order to ensure that the flow and water depth in the tank could be controlled based on operability, a centrifugal pump

was used to circulate the water in this flow tank. The water was pumped from a water storage mixing tank to the experimental tank through a pipeline. The maximum flow of the pump was 43.3 L/s, the head was 46 m, and a high-head, low-flow pump was used. The tank was equipped with a pump frequency conversion control cabinet and an E-MAG electromagnetic flowmeter to achieve accurate flow control. In this experiment, the flow range was 10~40 L/s and the water level range was 0.1~0.4 m. A water level gauge was placed on the outer wall of the glass tank. After the flow was stable, the water depth in the vegetation area was recorded, and then the flow and water level of each group were recorded.

(4) In the muddy water mixing tank, at the tail end of the tank, a mixing bar was constantly rotating during the test so that the sediment content in the test flow was maintained at a constant value of 16 kg/m$^3$. In addition, due to the muddy water flume test, three different silting thicknesses (i.e., 0, 6.5, and 11.5 cm) were used in the vegetation area to achieve the most realistic test results.

*2.2. Hydraulic Parameter Calculation*

The Reynolds and Froude numbers are two important hydraulic parameters. The Reynolds number is a dimensionless number that can be used to characterize fluid flow. The formula is $Re = VR/v$, where $V$ is flow velocity (m/s); $R$ is hydraulic radius (m); and $v$ is the kinematic viscosity coefficient. The Froude number is used to represent the state of water flow. When $Fr = 1$, it means that the water flow is affected equally by the inertial force and the gravity action, and it is a critical flow state. When $Fr > 1$, the inertial force is greater than gravity (the inertial force plays a dominant role), and the water flow is a jet stream, also known as a high flow state. When $Fr < 1$, the inertial force action is less than the gravity action (gravity plays a dominant role), and the water flow is slow, also known as a low flow state. The formula is $Fr = V/\sqrt{gh}$, where $V$ is flow velocity (m/s); $g$ is gravitational acceleration (m/s$^2$); and $h$ is test water depth (m). The calculated $Fr$ and $Re$ are shown in Table 1. It can be seen from Table 1 that the Froude numbers in the test were all less than 1, ranging from 0.11 to 0.19. This indicated that the test water flow was slow. The Reynolds number ranged from 9454 to 38062, which were all larger than the critical Reynolds number of an open channel. This indicated that the experimental water flow was turbulent. After the water pump was turned on and the flow meter reading stabilized, the water level scale was observed to determine when an approximately uniform flow had formed in the vegetation section. This was performed so that the relevant formula of constant uniform flow in the open channel could be used in the subsequent hydraulic calculation.

**Table 1.** Flow resistance test conditions.

| Silting Thickness (cm) | Q (L/s) | H (m) | Fr | Re |
|---|---|---|---|---|
| 0 | 7~43.06 | 0.118~0.31 | 0.1108~0.1594 | 9454~38,062 |
| 6.5 | 10~38.89 | 0.15~0.27 | 0.11~0.1771 | 12,376~37,023 |
| 11.5 | 10~35 | 0.15~0.24 | 0.11~0.1902 | 13,581~35,361 |

## 3. Experimental Results and Analyses

*3.1. Drag Force Coefficient of Vegetation*

When water flows through vegetation, vegetation has a drag effect on the water flow, resulting in a vegetation drag force. Usually, the vegetation drag force coefficient is expressed as a quantitative calculation of vegetation drag force. Water containing vegetation (Sections 1 and 2 in the flume) was taken as the control body to conduct the force analysis and obtain the expression of the vegetation drag force coefficient.

(1) Drag force of vegetation in a non-submerged state

Figure 3 shows the stress analysis of water in the control body section under the non-submerged vegetation state:

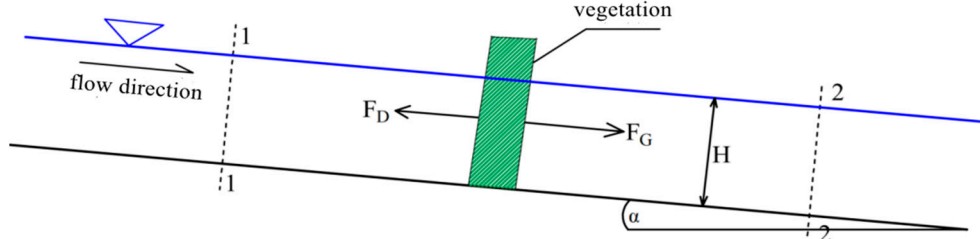

**Figure 3.** Vegetation stress in a non-submerged state.

Taking the water body in the control section (i.e., 1~2) as the research object, the force balance expression of the control body could be obtained as follows:

$$F_G = F_D + F_S \tag{1}$$

$$F_G = GSin\alpha = \rho_S g A J \tag{2}$$

$$F_D = C_D A_S N \frac{\rho_S U^2}{2} \tag{3}$$

where $F_G$ is the component controlling weight force along the direction of water flow, N; $F_D$ is the drag force of vegetation, N; $F_S$ is the resistance generated by the bottom boundary; $C_D$ is the drag force coefficient of vegetation, dimensionless and related to the shape of the object, angle of attack, and Reynolds number of incoming flow; $N$ is the number of plants in the control body; $A$ is the sectional area, $A = BH$, m²; $H$ is the water depth, m; $A_S$ is the cross-sectional area of vegetation in the direction of vertical water flow, which is related to the shape of vegetation, m²; $U$ is the average velocity of the section, m/s; and $\rho_S$ is the test flow density, kg/m³. Note that, because the sink boundary was smooth glass and the bed surface was PVC board, the side wall resistance and bed surface resistance were very small relative to the vegetation drag force and could be ignored by comparison; therefore, the drag force was considered equal to the component force of gravity along the direction of water flow. Formulas (1)~(3) were combined to obtain the following:

$$C_D A_S N \frac{\rho_S U^2}{2} = \rho_S g A J \tag{4}$$

The expression of vegetation drag force can be simplified as follows:

$$C_D = \frac{2gJ}{NU^2} \frac{A}{A_S} \tag{5}$$

(2) Vegetation drag force in a submerged state

Figure 4 shows the stress analysis of water in the control section when vegetation is submerged.

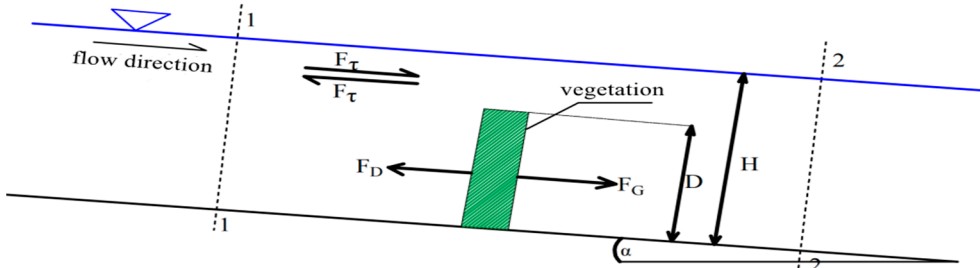

**Figure 4.** Stress of vegetation under submerged conditions.

When the water depth is greater than the vegetation height, the vegetation is considered submerged. In contrast to vegetation in a non-submerged state, shear stress occurs at the junction of vegetation and water flow when submerged, and there is no vegetation drag force in the water above the vegetation height.

$$F_\tau = \rho g B(H - D)J \tag{6}$$

The force analysis equation of the control body is:

$$F_G + F_\tau = F_D + F_S + F_\tau \tag{7}$$

Equations (2) and (3) can be combined to obtain the drag force coefficient of vegetation:

$$C_D = \frac{2gJ}{NU^2}\frac{BH}{A_S} \tag{8}$$

where $J$ is the water surface slope. In two vegetation-submerged states, the calculation methods for the cross-sectional area ($A_S$) of the vegetation upflow surface are different. If the vegetation model is approximately cylindrical, the calculation method of $A_S$ is relatively simple, i.e., $A_S = dH$ ($d$ is the diameter of the cylinder; $H$ is the water depth). Since the branches and leaves of the vegetation model in this test were mixed, $A_S$ was more difficult to calculate. Therefore, the simplified vegetation model was used to determine the cross-section area in flow. The vegetation model was simplified as two cylinders of different sizes: the diameter $d_1$ = 8 cm and height $h_1$ = 12 cm of the upper branches, and the diameter $d_2$ = 0.2 cm and height $h_2$ = 5 cm of the lower branches, obtained through measurement. $A_S$ is expressed as follows:

$$H \le h_2 \qquad\qquad A_S = d_2 H \tag{9}$$

$$h_2 < H \le (h_1 + h_2) \quad A_S = d_2 h_2 + \frac{(H - h_2)}{2}\left[2d_2 + \frac{d_1 - d_2}{h_1}(H - h_2)\right] \tag{10}$$

$$H > (h_1 + h_2) \qquad\qquad A_S = d_2 h_2 + \frac{1}{2}(d_1 + d_2)h_1 \tag{11}$$

Formula (9) was used to determine the cross-sectional area of vegetation ($A_S$). In order to obtain $C_D$, it is first necessary to obtain the number $N$ of the plants in the unit control body, which can be determined by Formula (12):

$$N = \frac{LB}{\Delta x \Delta y} \tag{12}$$

where $L$ and $B$ are the length and width of the unit control body, respectively, $m$, and $\Delta x$ and $\Delta y$ are the spacing between the plants, $m$.

The drag force coefficient of vegetation under different siltation conditions can be calculated using the above formula. Figure 5a–f shows the relationship of the vegetation drag force coefficient with the water flow Reynolds number (*Re*) and Froude number (*Fr*) under different vegetation deposition thicknesses. The figure shows that under different deposition conditions, the vegetation drag force coefficient ($C_D$) exhibited similar distribution trends. In all cases, $C_D$ decreased with increasing *Re* and *Fr*. This indicated that the greater the intensity of turbulence, the lower the drag force coefficients of vegetation. The amplitude tended to decrease, or become gentler, around *Re* = 15,000 and *Fr* = 0.13. However, if flow conditions are held constant, the $C_D$ increases with increasing sediment thickness in the vegetation area. These results show that the vegetation drag coefficient $C_D$ was related to both flow conditions (*Re* and *Fr*) and vegetation silting thickness. This was because, as the flow and water level increased, the branches and leaves of the plants were gradually submerged, which meant the turbulent intensity of the water flow in this

area was relatively large and $C_D$ was correspondingly reduced. Similarly, the $C_D$ increased with increasing vegetation siltation thickness because, as the siltation of the vegetation area increased, the degree of inundation under the same water flow condition increased. This resulted in more water flow through the vegetation's branch and leaf areas and led to a reduction in $C_D$. In addition, the water flow conditions (*Re* and *Fr*) had logarithmic relationships with $C_D$, and the fitting correlation coefficients were high in all conditions. The fitting formula and precision of the vegetation drag coefficient under various flow conditions are shown in Table 2.

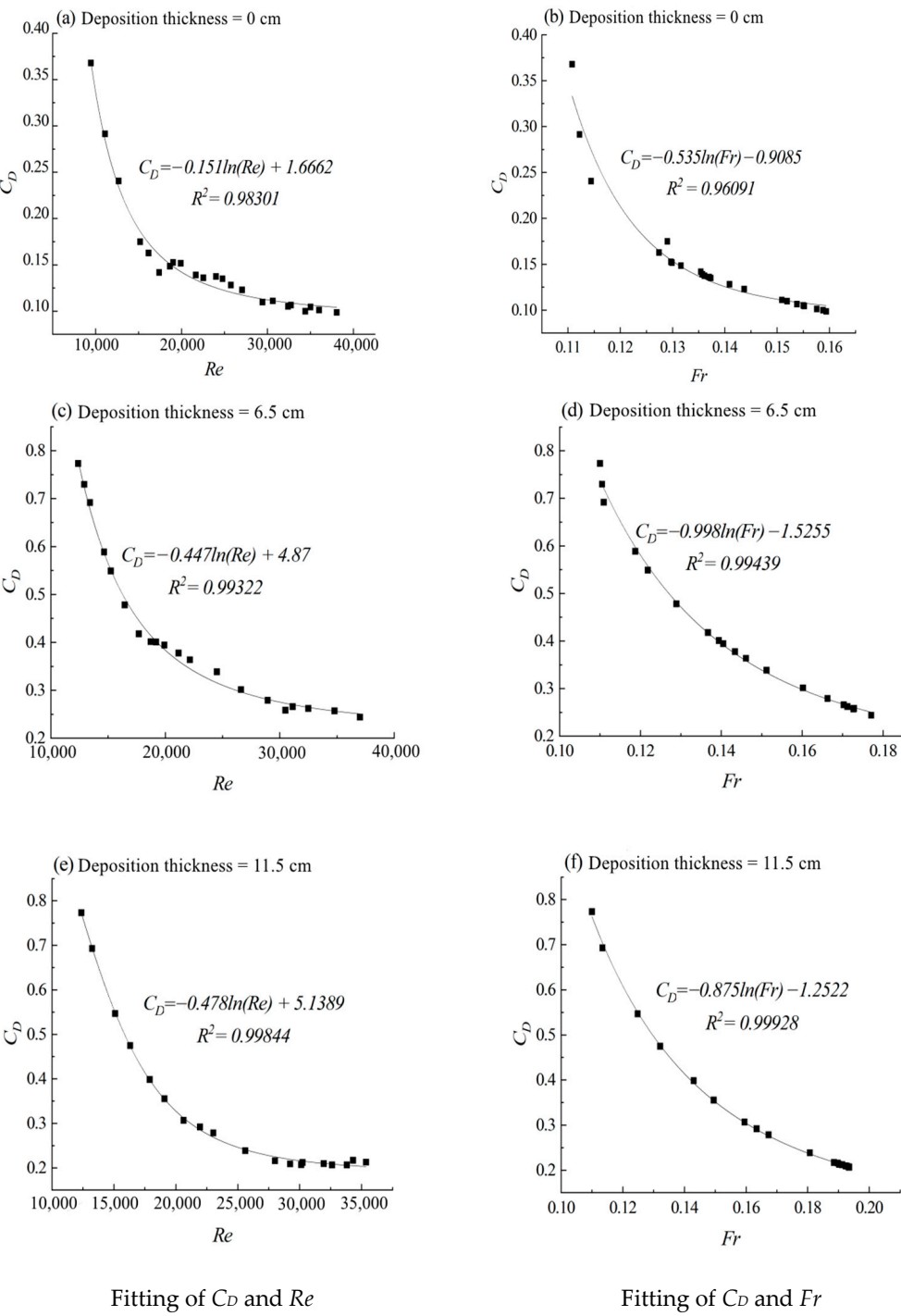

Fitting of $C_D$ and *Re*          Fitting of $C_D$ and *Fr*

**Figure 5.** Relationship between the drag coefficient of vegetation and Reynolds number and Froude number at different sediment thicknesses.

**Table 2.** Fitting formula of the drag force coefficients of vegetation under different deposition thicknesses.

| Silting Thickness | Re | $R^2$ | Fr | $R^2$ |
|---|---|---|---|---|
| 0 | $C_D = -0.151ln(Re) + 1.67$ | 0.98 | $C_D = -0.535ln(Fr) - 0.91$ | 0.96 |
| 6.5 | $C_D = -0.447ln(Re) + 4.87$ | 0.99 | $C_D = -0.998ln(Fr) - 1.53$ | 0.99 |
| 11.5 | $C_D = -0.478ln(Re) + 5.14$ | 0.99 | $C_D = -0.875ln(Fr) - 1.25$ | 0.99 |

According to the fitted formula, when $Re$ = 20,000 and $Fr$ = 0.12, $C_D$ changes with the change in deposition thickness, as shown in Figure 6.

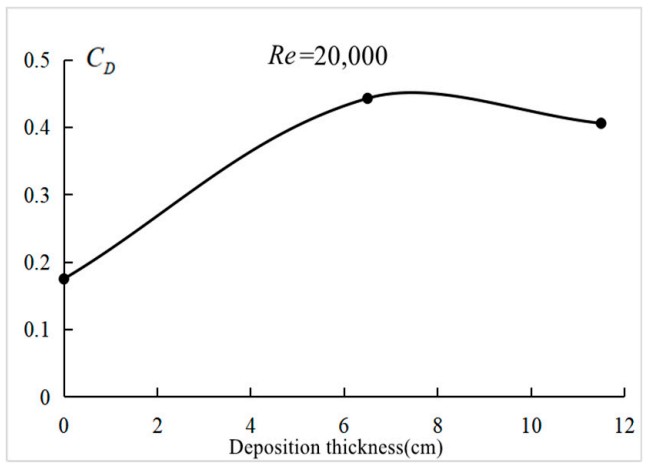

*Re* = 20,000 $C_D$ changes with deposition thickness

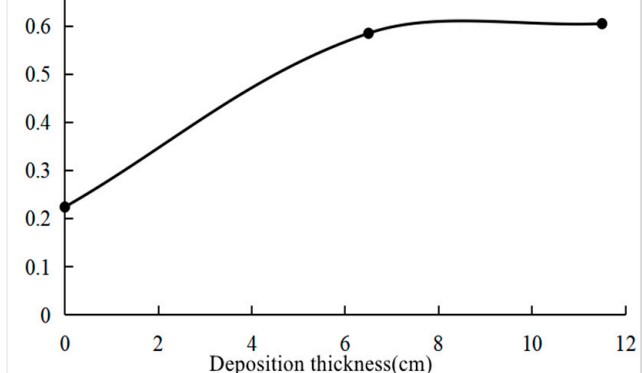

*Fr* = 0.12 $C_D$ changes with deposition thickness

**Figure 6.** When $Re$ = 20,000 and $Fr$ = 0.12, $C_D$ changes with deposition thickness.

### 3.2. Vegetation Roughness Coefficient

Manning's roughness coefficient is used to comprehensively reflect the resistance of pipes, channel walls, or rough water to water flow. It is also the most widely applied in resistance calculation, so this section analyzes how water conditions, sediment factors, and vegetation characteristics influence water flow using Manning's roughness coefficient. The analysis of muddy water containing vegetation was used to establish a fitting formula for vegetation roughness. Manning's roughness coefficient $n = \frac{1}{U}R^{2/3}J^{1/2}$, where $U$ is the average velocity of the section, m/s; $R$ is the hydraulic radius; and $J$ is the water surface slope. Since the water flow in the vegetation section was approximately uniform, the water surface slope was equal to the change due to the flume slope.

The resistances in this paper included both riverbank and riverbed resistances, among which riverbed resistance included the resistance generated by the riverbed surface and the vegetation in the river. This can be expressed by the formula:

$$\tau = \tau_w + \tau_b \tag{13}$$

where $\tau$ is the total shear stress in the river, and $\tau_W$ and $\tau_b$ are riparian resistance and riverbed resistance, respectively. The resistance segmentation method was used. The expressions of riparian and riverbed resistances are as follows:

$$\tau_w = \gamma R_w J \tag{14}$$

$$\tau_b = \gamma R_b J \tag{15}$$

where $R_w$ is the hydraulic radius relative to riverbank resistance; $R_b$ is the hydraulic radius relative to riverbed resistance; and $R_w = A_w/2h$, $R_b = A_b/B$.

According to the flow continuity equation and Manning's formula:

$$AU = A_w U_w + A_b U_b \tag{16}$$

$$U_w = \frac{1}{n_w} R_w^{2/3} J^{1/2} \tag{17}$$

$$U_b = \frac{1}{n_b} R_b^{2/3} J^{1/2} \tag{18}$$

where $n_w$ and $n_b$ represent riverbank roughness and riverbed roughness, respectively. Since the bed surface was a smooth organic PVC board in this experiment, its roughness can be ignored relative to the vegetation's roughness. $n_b$ is used to represent vegetation roughness, and it was assumed that the average velocity of each section in the river is equal, that is:

$$U = U_w = U_b \tag{19}$$

By combining Formulas (17)–(19), the following are obtained:

$$R_w = \left( \frac{n_w U}{J^{1/2}} \right)^{3/2} \tag{20}$$

$$R_b = h \left( 1 - 2 \frac{R_w}{B} \right) \tag{21}$$

Taking the edge wall roughness of the glass flume in the test as $n_w = 0.01$, then the vegetation roughness $n_b$ can be calculated from Equations (17)–(21). The relationships between the vegetation roughness coefficient $n_b$, water flow rates $Re$ and $Fr$, and vegetation sedimentation thickness were studied experimentally. Figure 7a–f shows how vegetation roughness and vegetation area deposition thickness affected the relationship of $n_b$ with $Re$ and $Fr$. The figure shows that, for all different vegetation siltation thicknesses, the vegetation roughness had negative correlations with $Re$ and $Fr$, i.e., $n_b$ decreased with increasing $Re$ and $Fr$. In terms of the effect of vegetation deposition thickness, when $Re$ and $Fr$ are held constant, the larger the deposition thickness, the larger the roughness, and vice versa. However, when $Re \geq 30{,}000$, vegetation roughness tended to stabilize around its minimum value under all siltation thicknesses. The relationship between the vegetation roughness coefficient and $Re$ and $Fr$ was established as a logarithmic function. This was true across all experimental conditions, as can be seen by the high fitting correlation coefficients ($R^2$) across treatments. The fitting formula and precision of the vegetation roughness coefficient under various working conditions are shown in Table 3.

**Table 3.** Fitting formula for vegetation roughness coefficients under different deposition thicknesses.

| Silting Thickness | $Re$ | $R^2$ | $Fr$ | $R^2$ |
|---|---|---|---|---|
| 0 | $n_b = -0.017ln(Re) + 0.26$ | 0.87 | $n_b = -0.063ln(Fr) - 0.04$ | 0.98 |
| 6.5 | $n_b = -0.034ln(Re) + 0.42$ | 0.98 | $n_b = -0.075ln(Fr) - 0.06$ | 0.99 |
| 11.5 | $n_b = -0.04ln(Re) + 0.48$ | 0.99 | $n_b = -0.073ln(Fr) - 0.06$ | 0.99 |

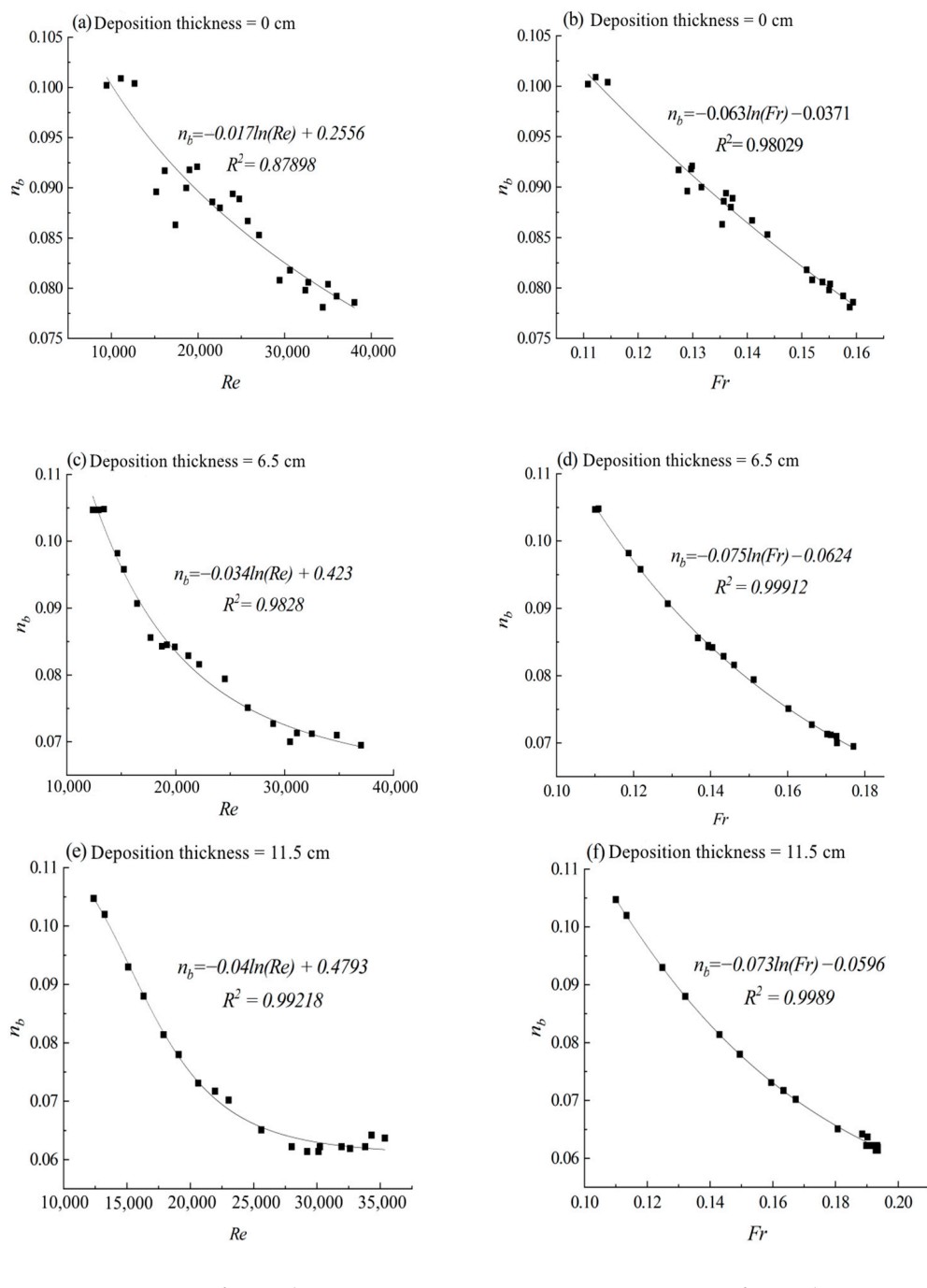

Fitting of $n_b$ and $Re$            Fitting of $n_b$ and $Fr$

**Figure 7.** Relationship between the vegetation roughness coefficient and the Reynolds and Froude numbers with different sediment thicknesses.

In previous calculations of vegetation resistance, vegetation roughness was generally used to reflect the resistance of the vegetation area. Wu et al. [20] used horse hair as the model vegetation in flume tests and studied the change in vegetation resistance with water depth, comparing submerged and non-submerged conditions. Their results showed that the roughness of non-submerged vegetation $n_b'$ decreased with increasing water depth to a certain vegetation height. When water depth was close to vegetation height, vegetation roughness $n_b'$ increased with water depth. As the water depth continued to increase after submerging the vegetation, $n_b'$ began to decrease and eventually became constant. The

relationships between vegetation roughness and vegetation drag force coefficient were therefore established:

Unsubmerged condition:

$$n_b = \left(\frac{R_b^{2/3}}{\sqrt{2g}}\right)\sqrt{C_D'} \qquad (22)$$

Submerged condition:

$$n_b = \left(\frac{D^{1/6}T^{1/2}}{\sqrt{2g}}\right)\sqrt{C_D'} \qquad (23)$$

where $R_b$ is the hydraulic radius corresponding to vegetation resistance, m; $T$ is vegetation height, m; and $C_D'$ is the drag force coefficient of vegetation. Cheng [24] studied the drag force of cylinders in water flow and proposed the concept of a generalized Reynolds number $C_D'$ and drag force coefficient $Re'$, wherein:

$$Re' = \frac{1+J}{1+80\lambda}Re \qquad (24)$$

$$C_D' = 11Re'^{-0.75} + 0.9\left[1 - exp\left(-\frac{1000}{Re'}\right)\right] + 1.2\left[1 - exp\left(-\left(\frac{Re'}{4500}\right)^{0.7}\right)\right] \qquad (25)$$

The expression of the vegetation resistance coefficient $n_b$ can be obtained by transforming the above-mentioned resistance formula:

$$n_b = \left(\frac{R_b^{2/3}}{\sqrt{2g}}\right)\sqrt{C_D'} \qquad (26)$$

where $R_b$ is the hydraulic radius corresponding to vegetation resistance, m, and $\lambda$ is the coefficient related to vegetation, $\lambda = 0.03$.

### 3.3. Darcy-Weisbach Resistance Coefficient

The experimental data were used to analyze the vegetation resistance in the beach area. According to the Darcy-Weisbach formula, the Darcy-Weisbach resistance coefficient can be derived, i.e., $f = 2gRJ/U^2$ where $R$ is hydraulic radius, $J$ is hydraulic slope, and $U$ is the average velocity of section. The relationship between Darcy-Weisbach resistance coefficient $f$ and water flow rates $Re$ and $Fr$ under different vegetation deposition conditions was established as shown in Figure 8a–f. The figure shows that the relationship between the Darcy-Weisbach resistance coefficient and flow $Re$ and $Fr$ in the vegetation area conformed to the Nikolaze curve under all different vegetation silting thickness conditions. With increasing vegetation area silting thickness, the resistance coefficient was adjusted to some extent due to the change in hydraulic conditions. When the $Re$ of the experimental flow was small ($Re < 20{,}000$), the drag coefficient varied greatly with $Re$. When the $Re$ continued to increase ($Re > 20{,}000$), the drag coefficient decreased with increasing $Re$. As the Reynolds number continued to increase, the resistance coefficient did not increase and eventually stabilized. Interestingly, the relationship between $Fr$ and the Darcy-Weisbach resistance coefficient in the vegetation area was slightly different from the above. Under all vegetation deposition conditions, with increasing $Fr$, the Darcy-Weisbach coefficient $f$ exhibited a nonlinear decrease. If flow and water level were held constant, the Darcy-Weisbach resistance coefficient would not change due to changes in vegetation siltation thickness. The logarithmic function correlation coefficient between water flow conditions and vegetation resistance coefficient was high, indicating reliable fitting accuracy. The fitting formula and precision of the vegetation roughness coefficient under various working conditions are shown in Table 4.

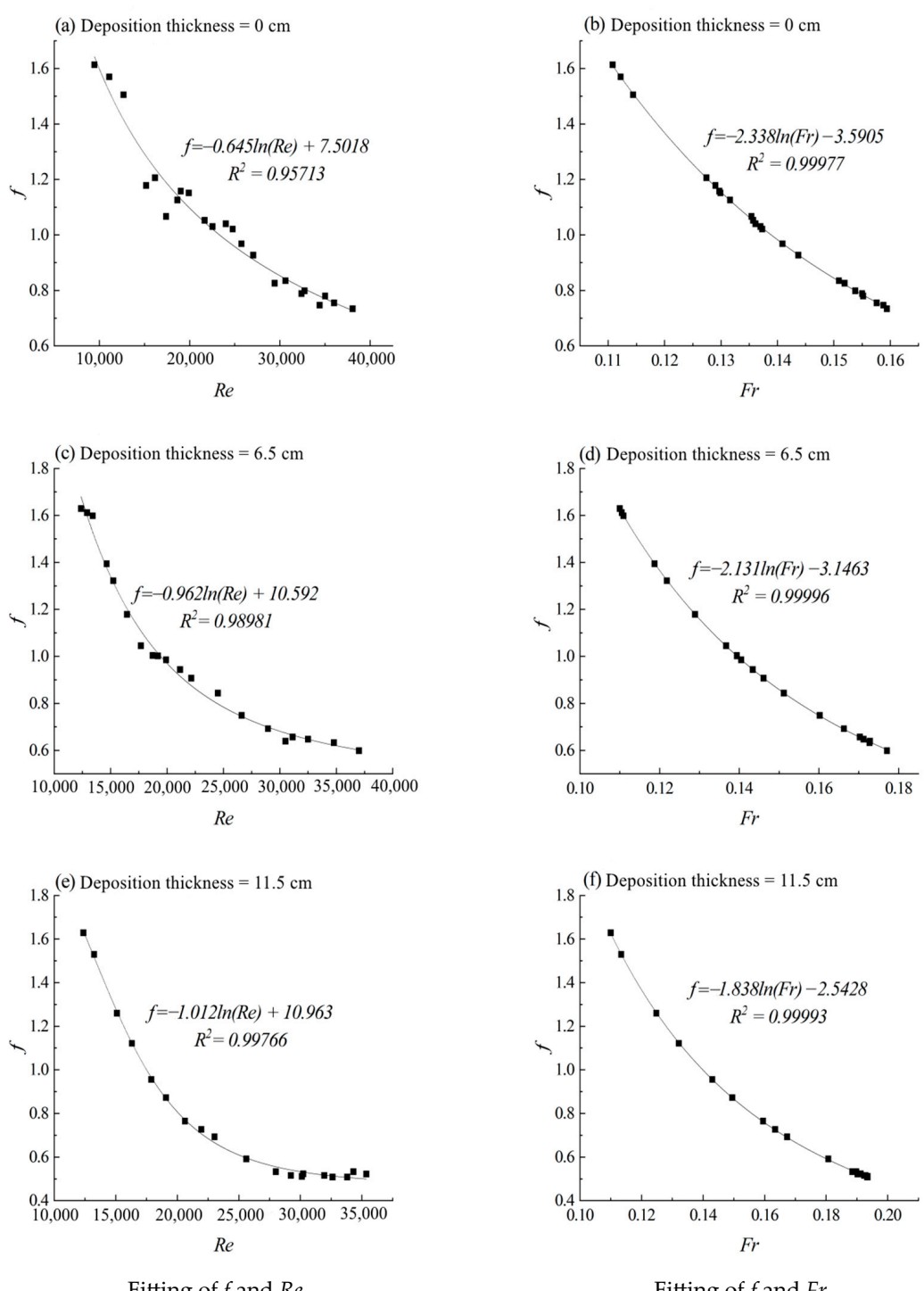

Fitting of *f* and *Re*                    Fitting of *f* and *Fr*

**Figure 8.** Relationship between the Darcy-Weisbach coefficient and Reynolds number and Froude number with different sediment thicknesses.

**Table 4.** Fitting formula for Darcy-Weisbach drag coefficient under different deposition thicknesses.

| Silting Thickness | Re | $R^2$ | Fr | $R^2$ |
|---|---|---|---|---|
| 0 | $f = -0.645ln(Re) + 7.50$ | 0.95 | $f = -2.338ln(Fr) - 3.59$ | 0.99 |
| 6.5 | $f = -0.962ln(Re) + 10.59$ | 0.98 | $f = -2.131ln(Fr) - 3.14$ | 0.99 |
| 11.5 | $f = -1.012ln(Re) + 10.96$ | 0.99 | $f = -1.838ln(Fr) - 2.54$ | 0.99 |

## 4. Conclusions

Taking maize-typical vegetation in the Lower Yellow River as the model vegetation, the influence of different sediment thicknesses on vegetation resistance characteristics in the muddy water area was simulated and studied by carrying out muddy water trough experiments with the same sand content and different silt thicknesses. The changes in drag force coefficient, roughness coefficient, and Darcy-Weisbach resistance coefficient with changes in the depth of sediment deposition were studied. This information has enriched our understanding of vegetation resistance in lower Yellow River beach areas and provided a scientific basis and reference for future studies of flood evolution in the lower Yellow River beach area. The main conclusions were as follows:

(1) For all vegetation deposition thicknesses (i.e., 0, 6.5, and 11.5 cm), the drag force coefficient of the vegetation decreased with increasing Reynolds and Froude numbers. In other words, with increased flow and water level, the drag force coefficient of vegetation decreased, and the decreased drag force variability around $Re = 15,000$ and $Fr = 0.13$ was indicative of gentler conditions. However, when the water flow conditions are the same, the vegetation drag coefficient increases with the increase in deposition thickness in the vegetation area.

(2) To analyze the resistance in river and beach areas, resistance is decomposed into riparian resistance and riverbed resistance, and the final calculation of vegetation roughness is obtained using the resistance segmentation method. With the same deposition thickness in the same vegetation area, the vegetation roughness coefficient was negatively correlated with Reynolds and Froude numbers. When $Re > 30,000$, under all deposition thicknesses, the vegetation roughness tended to stabilize near its minimum value. Logarithmic functions were established between the vegetation roughness coefficients and the Reynolds and Froude numbers of the different experimental water flow conditions, and all had high correlation coefficients.

(3) Based on the experimental data and the Darcy-Weisbach resistance formula, the expression of the Darcy-Weisbach resistance coefficient was deduced and analyzed. The results showed that when the Reynolds number was small ($Re < 20,000$), the Darcy-Weisbach resistance coefficient $f$ varied greatly with $Re$. When the $Re$ number was large ($Re > 20,000$), the Darcy-Weisbach resistance coefficient $f$ decreased with $Re$. With further increases in flow, or Reynolds number, the resistance coefficient tended to stabilize after reaching a certain value. Under the conditions of constant flow and water level, there was no significant correlation between the Darcy-Weisbach resistance coefficient and the thickness of vegetation deposition.

**Author Contributions:** Conceptualization, X.Z. and Y.Z.; methodology, H.W.; software, Z.B.; validation, Y.Z., H.W. and Z.X.; formal analysis, X.Z.; investigation, Y.Z.; resources, H.W.; data curation, X.Z.; writing—original draft preparation, Y.Z.; writing—review and editing, X.Z.; visualization, X.Z.; supervision, X.Z.; project administration, X.Z.; funding acquisition, X.Z. All authors have read and agreed to the published version of the manuscript.

**Funding:** This research was supported by the National Natural Science Foundation of China under contract No. 41930643, "Study on carbon and nitrogen process and its effect in the Lower Yellow River"; the National Natural Science Foundation of China, U2243220, "Efficient sand transport mechanism and critical regulation of sand content in the Lower Yellow River under changing circumstances"; and the Henan Province's key research, development, and promotion projects in 2022 (scientific and technological breakthroughs) under contract No. 222102320195. The authors are thankful to the anonymous reviewers for their valuable comments and suggestions to improve the quality of the paper.

**Data Availability Statement:** Data available in a publicly accessible repository that does not issue DOIs. Publicly available datasets were analyzed in this study.

**Conflicts of Interest:** The authors declare no conflict of interest.

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
