# Peer review of "Characteristics of Vegetation Resistance Variation in Muddy Water Flows"

_water, doi:10.3390/w15122238_

Round 1
Reviewer 1 Report
1. 'J' is not defined at its first occurrence.
2. Mistake in the text in line number 325
3. Not clear, how vegetation resistance is a function of the Froude number.
4. Muddy water is a visco-elastic fluid, however, the authors neglected it. This should be mentioned as an assumption.
English is OK.
Author Response
1.Defines J when it first appears.
2.The incorrect text has been fixed.
3.The publicity of vegetation resistance and Fround number fitting is presented in Table 4.
4.China‘s Yellow River is the river with the highest sand content in the world. The muddy water in this arcticle are assumed to be in the Yellow River Basin.

Reviewer 2 Report
General Comments:
The paper is based on the premise that muddy water (i.e., water with a relatively high sediment load) will affect the drag coefficient (CD), Manning’s roughness coefficient (n), and Darcy-Weisbach resistance coefficient (f), which are commonly used to characterize the drag force for clear water conditions. Yet this premise is not tested.
Specific Comments:
112: “MAEERIALS” is spelled incorrectly
140: 156 m3/h is an unusual way to state a flow rate; this would be equivalent to 43.3 l/s and is a small flow rate for a 46 m head. This head does not seem to correspond with Figure 1, which shows the water being drawn from a sump pit. Was a high-head low-flow pump used? This discrepancy needs an explanation.
156: “to represent the state of water flow” how about “to characterize the flow”
160: “This indicated that the test water flow was slow”, well, not really. It means the flow velocity was slow relative to the shallow water wave phase speed, that distinguishes critical from non-critical flow.
182: J is not defined
195: replace “integrated” with “combined” as integrated is really a calculus term
210: ditto re line 195
229: in the interest of consistency, the ‘x’s should be removed from equation 12. A multiply sign is not used elsewhere so it looks like a cross-product here, which it isn’t.
237: what does “This indicated what when the water flow condition became more and more unstable … “ Unstable how? A high Re doesn’t indicate more instability but rather increased turbulence as stated in line 245.
262: J is the bed slope (or water surface slope in the case of uniform flow); should be defined where it was first introduced (line 182).
264: “surface drop” should be replaced with “surface slope”
Closing Comments:
All tests were run with a sediment load of 16 kg/m3. What is left unexamined is the sensitivity of the results to sediment load. Moreover, how do the results in this paper compare to that for clear water (i.e., sediment load of 0 kg/m3)? To my mind this is a fundamental and critical flaw of this paper. Without any sense of the sensitivity of the results to sediment load, how does one apply these results to other sediment loads or clear water? The assertion that muddy water affects n, CD, and f, therefore, remains untested/unproven. Either the premise of the paper needs to change or this needs to be examined.
No concerns.
Author Response
112: Spell it correctly
140: 156 m3/h is equivalent to 43.3 l/s and a high-head low-flow pump was used .
156: Added how the Froud number characterizes the flow pattern of water.
160: According to the Froude number it is indeed slow flow.
182:Defines J when it first appears.
195: replace “integrated” with “combined” .
210: replace “integrated” with “combined” .
229: The ‘x’s is removed from equation 12.
237:This indicated that the greater the intensity of turbulence, the drag force coefficients of vegetation decreased and the drag force of vegetation decreased
262: :Defines J when it first appears.
264: “surface drop” is replaced with “surface slope”.
China's Yellow River is one of the rivers with the highest sand content in the world.The sand content selected in this paper is the annual average sand content.The study area in this paper assumes that there is no clear water situation in the Yellow River Basin.If it is clear water, there will be no siltation.

Round 2
Reviewer 2 Report
My primary concern with this paper was (and I quote from my previous review) ...
All tests were run with a sediment load of 16 kg/m3. What is left unexamined is the sensitivity of the results to sediment load. Moreover, how do the results in this paper compare to that for clear water (i.e., sediment load of 0 kg/m3)? To my mind this is a fundamental and critical flaw of this paper. Without any sense of the sensitivity of the results to sediment load, how does one apply these results to other sediment loads or clear water? The assertion that muddy water affects n, CD, and f, therefore, remains untested/unproven. Either the premise of the paper needs to change or this needs to be examined.
To which the authors replied ...
China's Yellow River is one of the rivers with the highest sand content in the world.The sand content selected in this paper is the annual average sand content.The study area in this paper assumes that there is no clear water situation in the Yellow River Basin.If it is clear water, there will be no siltation.
Reviewers follow up comments:
I understand that the Yellow River has a high sediment content. 16kg/m3 means that sediment makes up about 1.6% of the weight by volume, which is not very large. The authors state that sediment load affects n, CD, and f. But they still have not proven this. As this is the premise of the paper, I don't see how this paper can be published without proving this assertion. But I accept that if 16kg/m3 is a typically load sediment load in the Yellow River, than these results might be useful in that environment.
I would also like to note that I understand the importance of sedimentation on n, CD, and f. But this dependance introduces another factor that has not been properly discussed, and that is the time dependence of siltation. I expect that siltation depth has a stronger influence on n, CD, and f than the sediment load; these variables all have a time dependence in this study that has not been properly explored.
Author Response
Thank you for your hard review!Your opinion is quite right, because I did not express it clearly, you have a misunderstanding.Based on the flaws you said, I revised the premise of the thesis.In fact, I studied the effect of the same sand content and different sediment thickness on CD,n and f.It is not to study the effect of different sand content on CD,n and f.In fact, there are many people who have studied the effect of clear water on vegetation resistance.I did not study this, I did not express it clearly at first, and now I have revised the premise and conclusion.

Round 3
Reviewer 2 Report
The authors have reworked the introduction and conclusions to reflect that the paper is not about the effect of sediment concentration on n, CD, and f, but rather about the effect of sedimentation on these three (3) parameters. This is good. However, in my view the applicability of this research remains limited to the three (3) levels of deposition (0, 6.5, and 11.5 cm). What is missing is still missing from this paper:
i) how to interpolate n, CD, and f between these values, and
ii) some reference to the research that indicates how n, CD, and f vary with sediment load.
I think thought should be given to a diagram to address i). A response to ii) simply requires adding some references and text.
Please use track changes on this version of the paper so that one can readily see the changes made.
line 291: "The changes in drag force coefficient, roughness coefficient, and Darcy-Weisbach resistance coefficient with changes in sediment and water conditions were studied." This is not correct. The changes in drag force coefficient, roughness coefficient, and Darcy-Weisbach resistance coefficient with changes in the depth of sediment deposition were studied. [needs to be consistent with line 14 in the abstract]
The authors have modified the introduction and the conclusion. The grammar in those sections is now a bit rough in comparison to the rest of the manuscript.
Author Response
Thank you for your comments, your comments are especially valuable for this article.Based on your comments, I have made the following improvements:
i)I know you wanted to get the change in CD,n and f at different deposition thicknesses, so I plotted the plot under the conditions determined by Re and Fr according to the fitting formula, as shown in Figure 6. If I could interpolate in this way, I would complete all the interpolation results.
ii)I added some text on line 79 to prove that CD,n and f changes with sediment load.
Line 291 has been changed according to your comments.
